# Investigation of 8-Aza-7-Deaza Purine Nucleoside Derivatives

**DOI:** 10.3390/molecules24050983

**Published:** 2019-03-11

**Authors:** Hang Ren, Haoyun An, Jingchao Tao

**Affiliations:** 1College of Chemistry and Molecular Engineering, Zhengzhou University, 100 Science Avenue, Zhengzhou 450001, China; hang.ren@granlen.com; 2Zhengzhou Granlen PharmaTech, Ltd., 1300 Eastern Hanghai Road, Zhengzhou 450016, China

**Keywords:** glycosylation, 8-aza-7-deazapurine, nucleoside, pyrazolo[3,4-*d*]pyrimidine, 2D NMR, crystallographic analysis

## Abstract

Glycosylation of 6-amino-4-methoxy-1H-pyrazolo[3,4-*d*]pyrimidine and its iodo- and bromo- analogues with the protected ribofuranose and 2′-deoxyribofuranose under different conditions resulted in the synthesis of *N*^9^- and *N*^8^-glycosylated purine nucleosides. Five key intermediate nucleosides, having 6-methoxy, 7-iodo, and 2-bromo groups, were further derivatized to 23 final 8-aza-7-deazapurine nucleoside derivatives. The structures of *N*^9^- and *N*^8^-glycosylated products were assigned based on UV and NMR spectra. HMBC analysis of 2D NMR spectra and X-ray crystallographic studies of the representative compounds unambiguously verified the connection of ribose ring to *N*^9^- or *N*^8^-position of the purine ring. The anticancer activity of these new compounds was evaluated.

## 1. Introduction

8-Aza-7-deazapurines as well as their nucleosides exhibit a wide range of antiparasitic, antitumor, and antiviral activity [1,2]. A classical example of these biologically active analogues is allopurinol, which is an approved drug for the treatment of gout and it is also active against several *Leishmania* and trypanosome species [3]. 8-Aza-7-deazapurine-2,6-diamine and its riboside **5** are powerful purine nucleoside antagonists [4] (Figure 1), and 2′-deoxy-2′-β-fluoro-8-aza-7-deazapurine nucleosides display significant activity against HBV [5]. 8-Aza-7-deazapurines closely resemble the structure of purine with only C8 and N7 exchanged, which make their nucleosides excellent substrates of bio-enzymes [6,7] and ideal mimics of the canonical purine constituents of DNA or RNA. It has been demonstrated that the thermal stability of DNA duplexes increases significantly when 7-halogeno/alkynyl 8-aza-7-deazapurines replace purine [8]. Some *N*^8^-2′-deoxy-adenine analogues show ambiguous base pairing when incorporated into oligonucleotide duplexes opposite to the four canonical DNA-constituents [9]. Compounds **5** (pK = 5.8) and **11** (pK = 6.4) (Figure 1) are possibly protonated in neutral conditions when incorporated into RNA, thereby forming a stable mismatched base pair [10]. The RNA containing 7-deaza-8-azainosine was found to be a weaker toll-like receptor 8 activator than guanosine-containing RNA [11]. 7-Alkyne-8-aza-7-deaza nucleosides was used for azide-alkyne “Click” conjugation of DNA [12,13]. Furthermore, some other properties of such oligonucleotides have also been studied in drug discovery related fields [14,15].

Various 8-aza-7-deaza purine nucleosides have been synthesized in the past fewer years. However, there still remains a space for the design and development of new nucleoside-based therapeutics. We decided to investigate 8-aza-7-deaza purine nucleoside derivatives modified at the positions 6 and 7 of the purine ring. Herein, we report the synthesis of 23 modified 8-aza-7-deaza purine nucleoside derivatives, of which 11 are novel compounds, including 2-amino-6-substituted-ribonucleosides **1**–**3** and **5**–**13** (Figure 1), 2-amino-6-methoxy/oxo-7-substituted purine ribonucleosides **4** and **14**–**17** (Figure 1) and 2-amino-6-substituted 2′-deoxy nucleosides **18**–**23** (Figure 2). Different glycosylation protocols were studied for *N*^9^- and *N*^8^-glycosylation. In addition, the glycosylation sites and anomeric configuration of the newly synthesized compounds were investigated and assigned on the basis of ^1^H-NMR, ^13^C-NMR, 2D NMR, UV spectra, and X-ray crystallographic analysis of the representative compounds (see Appendix A). The anticancer activity of these compounds was also evaluated.

## 2. Results and Discussion

### 2.1. Synthesis of 8-Aza-7-deazapurine Ribo-Nucleosides

6-Amino-4-methoxy-1H-pyrazolo[3,4-*d*]pyrimidine (purine base **A**) [16], 6-amino-3-iodo-4-methoxy-1H-pyrazolo[3,4-*d*]pyrimidine (purine base **B**) [17], and 3,6-dibromo-1H-pyrazolo[3,4-*d*]pyrimidin-4(5H)-one (purine base **C**) [18] (Scheme 1) were made utilizing reported protocols, and selected as the heterocyclic bases of 8-aza-7-deazapurine nucleosides to be studied. These purines were chosen for a number of advantages including the switch in the nitrogen atom from position 7 to 8, which changes the binding mode and strength of purine nucleosides in the duplex nucleic acids and other biological systems, substantially altering the biological properties and application. The 2-amino-group on both 8-aza-7-deazapurine bases **A** and **B** does not need to be protected during the glycosylation process. The 6-methoxy group on purine bases **A** and **B** is selected to serve as a protecting group during glycosylation process. It is stable enough not to be cleaved in subsequent processes while active enough to be substituted by strong nucleophiles, therefore, converting to the desired corresponding 2-amino-8-aza-7-deazaguanosine derivatives when needed. The 6-oxo on purine base **C** does not need to be protected for glycosylation. The iodination of purine bases **A** to **B** adds high versatility for further derivatization on position 7 via Heck, Stille, Suzuki, Sonogashira, and related reactions. This will expand research in the related fields. To the best of our knowledge, the direct glycosylation of 2-amino-8-aza-7-deazapurine bases **A** and **B** with routinely used l-*O*-acetyl-2,3,5-tri-*O*-benzoyl-d-ribofuranose (ribose **I**) has not been reported.

A number of glycosylation protocols have been routinely used for the synthesis of different classes of nucleoside derivatives [10,19,20]. The silyl-Hilbert–Johnson glycosylation processes of silylated nucleobases with protected ribosides under Lewis acid conditions are very efficient for most cases. We glycosylated purine base **A** with ribose **I** (Scheme 1) under historic stannic chloride condition with no success. We then switched to the stronger promoter trimethylsilyl triflate (TMSOTf) and also explored silylating agents hexamethyldisilazane (HMDS) and *N,O*-bis(trimethylsilyl)acetamide (BSA) in acetonitrile, nitromethane and 1,2-dichloroethane. It was found that the HMDS-silylated purine base **A** was glycosylated with ribose **I** using TMSOTf in freshly distilled 1,2-dichloroethane overnight at room temperature giving the best result. The *N*^9^-glycosylated compound **24** (Scheme 1) was isolated in 30.3% yield as the major product, and only very trace amount of *N*^8^-glycosylated compound **25** was observed on TLC but could not be isolated.

We would like to develop a high-yield protocol that would lead to the large-scale synthesis of the key intermediates and diversified final 8-aza-7-deazaguanosine derivatives. We performed the reported protocol [10] using boron trifluoride ether solution as the Lewis acid promotor at an elevated temperature. The glycosylation of ribose **I** with purine base **A** in acetonitrile worked excellently. However, the *N*^8^-glycosylated product **25** was isolated in 83.3% high yield, and almost no *N*^9^-glycosylated compound **24** was observed under this condition (Scheme 1). The *N*^9^-glycosylated product **24** was the major product under room temperature glycosylation condition while the *N*^8^-glycosylated product **25** was the thermodynamically dominant product formed at an elevated temperature. Therefore, two different compounds were made under different conditions from the same materials.

Next, we set out to study the glycosylation of purine base **B**, which introduces a 7-iodo moiety that enables further derivatization. The glycosylation of iodo-purine base **B** with ribose **I** under both SnCl_4_ and TMSOTf conditions resulted in the complicated mixtures, and only trace amount of *N*^9^-glycosylated product **26** could be detected. We then utilized boron trifluoride-ether as the catalyst for this glycosylation in acetonitrile at room temperature. The desired *N*^9^-glycosylated product **26** was isolated in about 28% yield (Scheme 1). The purine base **C**, without 6-oxo-protection, was also glycosylated with ribose **I**, resulting in the protected nucleoside **27**. The two bromo-atoms on this molecule are expected to have different reactivity allowing for chemo-selective functionalization towards the synthesis of novel nucleoside derivatives.

The glycosylated products **24**, **25**, and **26** thus obtained were deprotected giving the corresponding 8-aza-7-deazapurine nucleosides **1**–**3**. These 6-methoxy derivatives were heated with ammonium hydroxide solution overnight providing the corresponding 2-amino-8-aza-7-deazaadenosine derivatives **5**, **8**, and **11** in excellent yields (Scheme 2). The 6-methoxy compounds **1**, **2**, and **3** were also treated with aqueous potassium hydroxide solution to undergo deprotection, yielding the corresponding 8-aza-7-deazaguanosine derivatives **7**, **9**, and **10**. Similarly, key intermediates **1** and **2** were heated with hydrazine solution to give the corresponding 6-hydrozino-purine nucleosides **6** and **12**. The 6-hydroxyamino purine nucleoside **13** was obtained by treatment of compound **2** with hydroxylamine at 60 °C. Therefore, the 6-methoxy group was effectively used as the protecting group during the glycosylation process, and it can be further substituted with various nucleophiles to form 6-modified purine nucleoside derivatives. When compound **27** was deprotected with NH_3_/MeOH at 120 °C, the bromo-atom at position 2 was substituted with NH_2_ group during the process, resulting in the corresponding 7-bromo-8-aza-7-deazaguanosine analogue **4**. Therefore, the bromo-atom at position 2 is more reactive against nucleophile than the bromo-atom at position 7.

The site of glycosylation and anomeric configuration of **24**, **25**, and **26** was assigned on the basis of ^1^H-NMR, ^13^C-NMR, 2D NMR, UV spectra, and X-ray crystallographic analysis of the representative compounds.

The nucleoside products **1** and **3** have very similar UV absorption compared to the corresponding free purine bases **A** and **B** (Table 1). This indicates that the ribofuranosyl group was glycosylated at the N9 position of purine bases **A** and **B** because the structure still maintains the same aromatic conjugate system [1]. However, the *N*^8^-glycosylated nucleosides showed very different UV spectra because the ribose-base bond connection dramatically altered the aromatic conjugate system. Careful UV spectral comparison of the *N*^9^- and *N*^8^-glycosylated nucleoside pairs shows the difference clearly. The *N*^8^-glycosylated products **2**, **10**, **11**, and **12** have UV maximum absorbances *λ*_max_ at longer wavelength than corresponding *N*^9^-glycosylated products **1**, **7**, **5**, and **6** (Figure 3).

We studied the 2D NMR (HMBC heteronuclear multiple bond correlation) of *N*^9^-glycosylated 8-aza-7-deazaguanosine **7** and *N*^8^-glycosyalted 8-aza-7-deazaguanosine derivatives **10** (Figure 4). 1′-H proton (5.88 ppm) on the ribose ring has a long-range coupling correlation with C4 (157.7 ppm) of the purine base for compound **7** while compound **10** does not have this coupling correlation. 1′-H (5.66 ppm) on the ribose ring has a long-range coupling correlation with C7 (128.2 ppm) of the purine base for compound **10** but compound **7** does not have the same coupling correlation. In addition, the 7-H on the purine ring of compound **10** is closer to the 1′-C of the ribose ring comparing to the distance of compound **7**. Therefore, the 7-H (8.525 ppm) on the purine ring of compound **10** has the long-range coupling correlation with 1′-C (94.057 ppm) of the ribose ring. However, compound **7** does not have the same coupling. Therefore, the 1′-C of the ribose ring is connected to N9 of the purine base on compound **7** (1′-H *vs* C4 coupling) while 1′-C of the ribose ring is connected to N8 of the purine base on the compound **10** (1′-H vs. C7 coupling and 7H *vs* 1′-C coupling). The data clearly further verified that compound **7** is the *N*^9^-glycosylated 8-aza-7-deazaguanosine, and compound **10** is the *N*^8^-glycosylated 8-aza-7-deazaguanosine derivative.

#### X-ray Crystallographic Study

Slow crystallization of compounds **7** and **8** from water gave X-ray quality crystals. A suitable crystal was selected and mounted on a Xcalibur, Eos, Gemini diffractometer. The crystal was kept at 293(2) K during data collection. Using Olex2, the structure was solved with the ShelXS structure solution program using Direct Methods and refined with the ShelXL refinement package using Least Squares minimization. The resulted crystal structures are shown in Figure 5 and Figure 6. The results confirmed the β-D-anomeric configuration and the site of glycosylation as N9. Consequently, the structures of the corresponding products **7** and **8** were further verified.

Furthermore, the iodinated intermediate **3** is a valuable starting point for the introduction of aromatic or alkynyl substituents by the Pd catalyzed cross-coupling reaction. The 7-iodine intermediate **3** was coupled with 2-(tributylstannyl)furan catalyzed by bis(triphenylphosphine) palladium(II) chloride at 90 °C in DMF providing derivative **14** in 90% yield. Compound **3** was reacted with phenylboronic acid catalyzed by tetrakis(triphenylphosphine)palladium (Pd(PPh_3_)_4_) in DME–water (2:1) mixture resulting in the desired 7-substituted product **15** in 80% yield. The 7-alkynylated products **16** and **17** were obtained in high yields by Sonogashira coupling reaction of **3** with alkynyl reagents (Scheme 3). The 8-aza-7-deazaguanine derivatives having 7-iodo (compound **9**) and 7-bromo (compound **4**) gave extremely low yields under Pd-catalyzed cross-coupling reaction conditions. Therefore, protection of 6-oxo group by methoxy group is required to activate the 7-halo atom for C-C coupling reactions.

### 2.2. Synthesis of 8-Aza-7-deaza-2′-deoxy Purine Nucleosides

2′-Deoxy-ribofuranosyl halide was utilized to glycosylate with various 7-dezaz-, 8-aza-7-deaza- and other purine related heterocyclic bases [21,22,23,24]. We did the glycosylation of purine base **A** with 2-deoxy-3,5-di-*O*-(*p*-toluoy1)-α-d*-erythro*-pentafuranosyl chloride (ribose **II**) [24,25] under the typical potassium conditions in the presence of cryptand TDA-1 catalyst. The desired *N*^9^-β-glycosylated compound **29** was obtained in 52% yield, and the α-anomer **28** was also isolated in 11% yield (Scheme 4) [16]. Surprisingly, *N*^8^-glycosylated compound was not observed. Deprotection of compounds **28** and **29** was accomplished with sodium methoxide at ambient temperature to give product **18** and **19** in 89% and 91% yields, respectively (Scheme 4). The 6-methoxy group of compound **19** was reacted with sodium hydroxide, ammonium hydroxide, hydrazine, and hydroxylamine as described above giving the corresponding 8-aza-7-deaza-2′-deoxy-purine nucleoside derivatives **20**–**23** in good yield.

## 3. Biological Evaluation

Newly synthesized compounds were tested for inhibitory activity against human lung carcinoma cell line A549 and the human breast cancer cell line MDA-MB-231 (Table 2). We found that the 7-iodine substituted derivative **8** showed the best inhibitory activity against A549 with an IC_50_ value of 7.68 μM. Compounds **14** and **16** showed slightly better activity than other modified derivatives, which did not show detectable activity against the tested tumor cell lines. By comparing the activity of tested analogs with different substituents, it suggested that electron-donating group at 6-position of the base may improve the activity. Further biological evaluation of these modified nucleosides is in progress and will be reported in due course.

## 4. Materials and Methods

### 4.1. Materials and Reagents

Starting materials and reagents were obtained from commercial suppliers and were used without further purification unless otherwise stated. ^1^H NMR spectra were obtained with a Bruker Avance 400 spectrometer using DMSO-*d*_6_ or CDCl_3_ (ppm) downfield with respect to an internal δ as solvents. Chemical shifts are reported as standard of tetramethylsilane (TMS). Product purity was tested by an Agilent 1260 analytical HPLC system. LC-MS spectra were measured on an Agilent 6120 LC–MS spectrometer. High-resolution mass spectra were obtained on a Micro-Q-TOF mass spectrometer. Single-crystal structure was determined by a Xcalibur, Eos, Gemini X-ray single crystal diffractometer. TLC was performed on silica gel GF254. Flash chromatography was performed on silica gel 200–300 mesh (Yantai Silica Gel Co. LTD). 6-Amino-4-methoxy-1H-pyrazolo[3–*d*]pyrimidine (purine base **A**), 6-amino-3-iodo-4-methoxy-1H-pyrazolo[3,4-*d*]pyrimidine (purine base **B**) and 3,6-dibromo-1H-pyrazolo[3,4-*d*]pyrimidin-4(5H)-one (purine base **C**) were prepared following literature procedures [16,17,18]. The synthesis of *N*-(2-propynyl)-2,2,2-trifluoroacetamide was accomplished from propargylamine according to the reported protocol [26].

### 4.2. Chemical Synthesis

#### 4.2.1. Procedure for Preparation of Glycosylated Products **24**–**26** and Key Intermediate **3**

*Synthesis of 6-amino-4-methoxy-1-(2,3,5-tri-O-benzoyl-β-d-ribofuranosyl)-1H-pyrazolo[3,4-d]pyrimidine* (**24**): A suspension of 6-amino-4-methoxy-1H-pyrazolo[3.4-*d*]pyrimidine (purine base **A**, 9.0 g, 54.5 mmol, 1.0 eq) and a catalytic amount of ammonium sulfate in hexamethyldisilazane (HMDS, 150 mL) was refluxed for 6 h. The excess hexamethyldisilazane was removed by evaporation under reduced pressure, and the residue was dissolved in 1,2-dichloroethane (200 mL). l-*O*-Acetyl-2,3,5-tri-*O*-benzoyl-d-ribofuranose (ribose **I**) (35.7 g, 70.8 mmol, 1.3 eq) was added at room temperature. The reaction mixture was cooled to 0 °C, and trimethylsilyl trifluoromethanesulfonate (TMSOTf, 29.6 mL, 163.5 mmol, 3.0 eq) was added dropwise for 30 min with stirring. The reaction mixture was stirred at room temperature overnight. Upon completion of the reaction as monitored by TLC, the mixture was diluted with dichloromethane (200 mL) and washed with saturated sodium bicarbonate solution. The aqueous layer was extracted with dichloromethane. The combined organic layer was dried over anhydrous sodium sulfate and concentrated under reduced pressure. The resulting residue was purified by column chromatography to afford 10.0 g main product *N*^9^ β-isomer **24** as a white solid in 30.3% yield with an HPLC purity of 96%; *R_f_* = 0.3 (petroleum ether–ethyl acetate = 1:1).

*Synthesis of 6-amino-4-methoxy-2-(2,3,5-tri-O-benzoyl-β-d-ribofuranosyl)-2H-pyrazolo[3,4-d]pyrimidine* (**25**): 6-Amino-4-methoxy-1H-pyrazolo[3,4-*d*]pyrimidine (purine base A; 8.0 g, 48.4 mmol, 1.0 eq) was suspended in 200 mL dry acetonitrile and 1-*O*-acetyl-2,3,5-tri-*O*-benzoyl-d-ribofuranose (ribose **I**) (36.6 g, 72.6 mmol, 1.5 eq) was added. The reaction mixture was heated to reflux at 95 °C, and the freshly distilled BF_3_·OEt_2_ (12.2 mL, 96.8 mmol, 2.0 eq) was then added with stirring. The reaction mixture became clear immediately and then slowly became dark. The mixture was stirred at this temperature for 15 min. Upon the completion of the reaction as monitored by TLC, the reaction mixture was cooled to room temperature and concentrated under reduced pressure. The resulting residue was purified by column chromatography to afford 25.0 g *N*^8^-product **25** as a white solid in 83.3% yield with an HPLC purity of 96.8%. *R_f_* = 0.6 (dichloromethane–methanol = 30:1). UV–vis (MeOH) *λ*_max_: 225 nm; ESI-MS *m*/*z*: 610.6 [M + H]^+^.

*Synthesis of 6-amino-3-iodo-4-methoxy-1-(2,3,5-tri-O-benzoyl-β-d-ribofuranosyl)-1H-pyrazolo[3,4-d]-pyrimidine* (**26**) *and 6-amino-3-iodo-4-methoxy-1-(β-d-ribofuranosyl)-1H-pyrazolo[3,4-d]pyrimidine* (**3**): 6-Amino-3-iodo-4-methoxy-1H-pyrazolo[3,4-d]pyrimidine (purine base **B**, 5.0 g, 17.2 mmol, 1.0 eq) and 1-O-acetyl-2,3,5-tri-O-benzoyl-d-ribofuranose (ribose **I**) (13.0 g, 25.8 mmol, 1.5 eq) were suspended in 200 mL of anhydrous acetonitrile. The freshly distilled BF_3_·OEt_2_ solution (3.2 mL, 34.4 mmol, 2.0 eq) was added in 30 min with stirring at room temperature, and the reaction mixture was stirred at the same temperature for 5 h. Upon the completion of the reaction as monitored by TLC, the reaction mixture was poured into 500 mL of saturated sodium bicarbonate solution and extracted with ethyl acetate. The organic phase was separated, and the aqueous phase was extracted with ethyl acetate. The combined organic phases were dried over anhydrous sodium sulfate. The drying agent was filtered off, and the filtrate was concentrated under reduced pressure to afford crude product **26**, which was used directly without further purification. R*_f_* = 0.2 (dichloromethane–methanol = 30:1). The crude compound **26** was dissolved in 100 mL of MeOH and 10 mL of THF solution. Then sodium methoxide (2.8 g, 51.6 mmol, 3.0 eq) was added, and the mixture was stirred at room temperature for 5 h. Upon completion of the reaction as monitored by TLC, the mixture was neutralized with 2 N HCl solution and evaporated under reduced pressure. The resulting residue was purified by column chromatography to afford 1.8 g desired product **3** as a white solid in 25% overall yield with an HPLC purity of 98.5%. R_f_ = 0.2 (dichloromethane–methanol = 10:1). UV–vis (MeOH) *λ*_max_: 232, 278 nm; ^1^H NMR (DMSO-d6, 400 MHz) δ 7.02 (s, 2H, NH_2_), 5.90 (d, *J* = 4.8 Hz, 1H, 1′-H), 5.33 (d, 1H, *J* = 6.0 Hz, 2′-OH), 5.08 (d, *J* = 5.2 Hz, 1H, 3′-OH), 4.74 (t, *J* = 5.6 Hz, 1H, 5′-OH), 4.45–4.55 (m, 1H, 2′-H), 4.08–4.15 (m, 1H, 3′-H), 3.09 (s, 3H, CH_3_), 3.80–3.87 (m, 1H, 4′-H), 3.38–3.58 (m, 2H, CH_2_); ESI-MS *m*/*z*: 424.3 [M + H]^+^, 446.2 [M + Na]^+^.

#### 4.2.2. Procedure for Preparation of Glycosylated Product **27** and Compound **4**

*Synthesis of**3,6-dibromo-4-hydro-1-(2,3,5-tri-O-benzoyl-β-d-ribofuranosyl)-1H-pyrazolo[3,4-d]pyrimidine* (**27**): 3,6-Dibromo-1H-pyrazolo[3,4-*d*]pyrimidin-4(5H)-one (purine base **C**; 14.2 g, 48.4 mmol, 1.0 eq) was suspended in 300 mL dry acetonitrile and 1-*O*-acetyl-2,3,5-tri-*O*-benzoyl-d-ribofuranose (ribose **I**) (36.6 g, 72.6 mmol, 1.5 eq) was added. The reaction mixture was heated to reflux at 95 °C, and the freshly distilled BF_3_·OEt_2_ (12.2 mL, 96.8 mmol, 2.0 eq) was then added with stirring. The reaction mixture became clear immediately and then slowly became dark. The mixture was kept stirred at this temperature for 20 min. Upon the completion of the reaction as monitored by TLC, the reaction mixture was cooled to room temperature and concentrated under reduced pressure. The resulting residue was purified by column chromatography to afford 33 g product **27** as a white solid in 92% yield with an HPLC purity of 98%. *R_f_* = 0.5 (dichloromethane–methanol = 20:1). UV–vis (MeOH) *λ*_max_: 231, 275 nm; ^1^H NMR (DMSO-*d6*, 400 MHz) δ 8.06–8.16 (m, 2H, Ar-H), 7.91–8.01 (m, 4H, Ar-H), 7.51–7.61 (m, 3H, Ar-H), 7.34–7.46 (m, 6H, Ar-H), 6.62 (d, *J* = 2.8 Hz, 1H, 1′-H), 6.27–6.36 (m, 1H, 2′-H), 6.17–6.26 (m, 1H, 3′-H), 4.72–4.89 (m, 2H, 4′-H, CH_2_), 4.57–4.70 (m, 1H, CH_2_); ESI-MS *m*/*z*: 761.0 [M + Na]^+^.

*Synthesis of 6-amino-3-bromo-1,5-dihydro-1-(β-d-ribofuranosyl)-4H-pyrazolo[3,4-d]pyrimidin-4-one* (**4**): Compound **27** (9.0 g, 12.2 mmol) was dissolved in NH_3_/MeOH (300 mL, saturated at 0 °C) in high pressure reactor, and the reaction mixture was heated at 120 °C overnight. Upon completion of the reaction as monitored by TLC, the reaction solvent was evaporated under reduced pressure. The resulting precipitate was filtered off and recrystallized from water to afford 2.5 g compound **4** as a white solid in 56% yield with an HPLC purity of 98%. R*_f_* = 0.3 (dichloromethane–methanol = 1:1). UV–vis (MeOH) *λ*_max_: 223, 255 nm; ESI-MS *m*/*z*: 363 [M + H]^+^.

#### 4.2.3. Procedure for Preparation of Key Intermediates **1** and **2**

*Synthesis of 6-amino-4-methoxy-1-(β-d-ribofuranosyl)-1H-pyrazolo[3,4-d]pyrimidine* (**1**): Compound **24** (14.5 g, 23.8 mmol) was dissolved in a mixture of methanol (200 mL) and THF (50 mL). Sodium methoxide (3.9 g, 71.4 mmol, 3.0 eq) was then added, and the mixture was stirred at room temperature. Upon completion of the reaction as monitored by TLC, the mixture was neutralized with 2 N HCl solution. Then the reaction mixture was evaporated under reduced pressure, and the resulting residue was purified by column chromatography to afford 6.2 g desired product **1** as a white solid in 87% yield with an HPLC purity of 98.2%. R*_f_* = 0.2 (dichloromethane–methanol = 10:1). UV–vis (MeOH) *λ*_max_: 222, 252, 276 nm; ^1^H NMR (DMSO-*d6*, 400 MHz) δ 7.90 (s, 1H, Ar-H), 6.86 (s, 2H, NH_2_), 5.97 (d, *J* = 4.8 Hz, 1H, 1′-H), 5.32 (d, *J* = 5.6 Hz, 1H, 2′-OH), 5.05 (d, *J* = 5.6 Hz, 1H, 3′-OH), 4.74 (t, *J* = 5.6 Hz, 1H, 5′-OH), 4.46–4.56 (m, 1H, 2′-H), 4.13–4.21 (m, 1H, 3′-H), 3.97 (s, 3H, CH_3_), 3.82–3.89 (m, 1H, 4′-H), 3.49–3.60 (m, 1H, 5′-H), 3.36–3.47 (m, 1H, 5′-H); ESI-MS *m*/*z*: 298.1 [M + H]^+^, 320.1 [M + Na]^+^.

*Synthesis of 6-amino-4-methoxy-2-(β-d-ribofuranosyl)-2H-pyrazolo[3,4-d]pyrimidine* (**2**): As described for **1**, 6-amino-4-methoxy-2-(2,3,5-tri-*O*-benzoyl-β-d-ribofuranosyl)-2H-pyrazolo[3,4-*d*]pyrimidine **25** (15 g, 24.6 mmol) was converted to 6.2 g of compound **2** as a white solid in 85% yield with an HPLC purity of 97.2%. R*_f_* = 0.15 (dichloromethane–methanol = 10:1). UV–vis (MeOH) *λ*_max_: 222, 266, 296 nm; ^1^H NMR (DMSO-*d6*, 400 MHz) δ 8.53 (s, 1H, Ar-H), 6.46 (s, 2H, NH_2_), 5.75 (d, *J* = 3.2 Hz, 1H, 1′-H), 4.28–4.34 (m, 1H, 2′-H), 4.15 (t, *J* = 5.2 Hz, 1H, 3′-H), 3.92–4.01 (m, 4H, 4′-H, CH_3_), 3.66–3.72 (m, 1H, 5′-H), 3.51–3.57 (m, 1H, 5′-H); ESI-MS *m*/*z*: 297.9 [M + H]^+^. HRMS calcd for C_11_H_16_N_5_O_5_ 298.1151 [M + H]^+^, found 298.1153.

#### 4.2.4. Procedure for Preparation of Target Compounds **5**–**17**

*Synthesis of 4,6-diamino-1-(β-d-ribofuranosyl)-1H-pyrazolo[3,4-d]pyrimidine* (**5**): Compound **1** (2.0 g, 6.7 mmol) was dissolved in 25% ammonium hydroxide (40 mL) in high pressure reactor, and the reaction mixture was heated at 80 °C overnight. Upon completion of the reaction as monitored by TLC, the reaction solvent was evaporated under reduced pressure. The resulting precipitate was filtered off and recrystallized from water to afford 900 mg of compound **5**. The filtrate was concentrated under reduced pressure, and the resulting residue was purified by column chromatography to afford 520 mg more desired product **5** as a white solid. Totally, 1.42 g desired product was obtained in 75% yield with an HPLC purity of 96.8%. R*_f_* = 0.2 (dichloromethane–methanol = 4:1). UV–vis (MeOH) *λ*_max_: 223, 254, 275 nm; ^1^H NMR (DMSO-*d6*, 400 MHz) δ 7.97 (s, 1H, Ar-H), 7.38 (br, 2H, NH_2_), 6.34 (s, 2H, NH_2_), 5.91 (d, *J* = 4.4 Hz, 1H, 1′-H), 5.21–5.38 (br, 1H, 2′-OH), 5.02–5.20 (br, 1H, 3′-OH), 4.45–4.55 (m, 1H, 2′-H), 4.08–4.20 (m, 1H, 3′-H), 3.80–3.97 (m, 1H, 4′-H), 3.38–3.58 (m, 2H, CH_2_); ESI-MS *m*/*z*: 283.1 [M + H]^+^.

*Synthesis of 6-amino-4-hydrazino-1-(β-d-ribofuranosyl)-1H-pyrazolo[3,4-d]-pyrimidine* (**6**): Compound **1** (2.0 g, 6.7 mmol) was dissolved in 80% hydrazine hydrate solution (30 mL), and the reaction mixture was heated at 70 °C for 3 h. Upon completion of the reaction as monitored by TLC, the reaction solvent was evaporated under reduced pressure. The resulting precipitate was filtered off and recrystallized from water to afford 850 mg compound **6**. The filtrate was concentrated under reduced pressure, and the resulting residue was purified by column chromatography to afford 550 mg desired product **6** as a white solid. Totally, 1.4 g desired product was obtained in 72% yield with an HPLC purity of 96%. R*_f_* = 0.1 (dichloromethane–methanol = 3:1). UV–vis (MeOH) *λ*_max_: 225, 275 nm; ^1^H NMR (DMSO-*d6*, 400 MHz) δ 8.98, 8.50 (br, 1H, NH), 8.10, 7.86 (br, 1H, NH), 6.23 (br, 1H, NH), 6.01 (br, 1H, NH), 5.93 (d, *J* = 4.4 Hz, 1H, 1′-H), 5.25 (d, *J* = 5.6 Hz, 1H, 2′-OH), 5.01 (d, *J* = 4.0 Hz, 1H, 3′-OH), 4.78–4.96(m, 1H, 5′-OH), 4.55–4.74 (m, 1H, NH), 4.40–4.54 (m, 1H, 2′-H), 4.07–4.24 (m, 1H, 3′-H), 3.76–3.92 (m, 1H, 4′-H), 3.36–3.58 (m, 2H, 5′-H); ESI-MS *m*/*z*: 298.1 [M + H]^+^. HRMS calcd for C_10_H_16_N_7_O_4_ 298.1264 [M + H]^+^, found 298.1265.

*Synthesis of 6-amino-1,5-dihydro-1-(β-d-ribofuranosyl)-4H-pyrazolo[3,4-d]pyrimidin-4-one* (**7**): Compound **1** (2.0 g, 6.7 mmol) was dissolved in 2N NaOH solution (50 mL). The reaction mixture was stirred at room temperature overnight. Upon completion of the reaction as monitored by TLC, the reaction mixture was neutralized with 6 N HCl under cooling condition and then concentrated under reduced pressure. The resulting precipitate was filtered off and recrystallized from water to yielding 1.2 g of compound **7** as a white solid in 62% yield with an HPLC purity of 96.3%. R*_f_* = 0.2 (dichloromethane–methanol = 4:1). UV–vis (MeOH) *λ*_max_: 219, 254 nm; ^1^H NMR (DMSO-*d6*, 400 MHz) δ 10.61 (s, 1H, Ar-NH), 7.85 (s, 1H, Ar-H), 6.70 (s, 2H, NH_2_), 5.88 (d, *J* = 4.4 Hz, 1H, 1′-H), 5.31 (d, 1H, *J* = 5.6 Hz, 2′-OH), 5.05 (d, *J* = 4.2 Hz, 1H, 3′-OH), 4.73 (t, *J* = 5.6 Hz, 1H, 5′-OH), 4.42–4.52 (m, 1H, 2′-H), 4.10–4.20 (m, 1H, 3′-H), 3.80–3.90 (m, 1H, 4′-H), 3.38–3.62 (m, 2H, CH_2_); ^13^C NMR (100 MHz, DMSO-*d*6) *δ*: 157.7 (C(4)), 155.9 (C(6)), 154.9 (C(2)), 135.4 (C(7)), 99.6 (C(5)), 87.5 (C(1′)), 84.8 (C(4′)), 73.0 (C(2′)), 70.8 (C(3′)), 62.4 (C(5′)); ESI-MS *m*/*z*: 284.1 [M + H]^+^.

*Synthesis of 4,6-diamino-3-iodo-1-(β-d-ribofuranosyl)-1H-pyrazolo[3,4-d]pyrimidine* (**8**): As described for **5**, 6-amino-3-iodo-4-methoxy-1-(β-d-ribofuranosyl)-1H-pyrazolo[3,4-*d*]pyrimidine **3** (1.3 g, 3.0 mmol) was converted to 860 mg of compound **8** as a white solid in 69% yield with an HPLC purity of 95.3%. R*_f_* = 0.3 (dichloromethane–methanol = 6:1). UV–vis (MeOH) *λ*_max_: 229, 263, 278 nm; ^1^H NMR (DMSO-*d6*, 400 MHz) δ 6.37 (s, 2H, NH_2_), 5.85 (d, *J* = 4.8 Hz, 1H, 1′-H), 5.29 (d, 1H, *J* = 5.6 Hz, 2′-OH), 5.04 (d, *J* = 4.2 Hz, 1H, 3′-OH), 4.78 (t, *J* = 5.6 Hz, 1H, 5′-OH), 4.43–4.54 (m, 1H, 2′-H), 4.06–4.16 (m, 1H, 3′-H), 3.80–3.86 (m, 1H, 4′-H), 3.37–3.60 (m, 2H, CH_2_); ESI-MS *m*/*z*: 409.0 [M + H]^+^, 431.0 [M + Na]^+^.

*Synthesis of 6-amino-3-iodo-1,5-dihydro-1-(β-d-ribofuranosyl)-4H-pyrazolo[3,4-d]pyrimidin-4-one* (**9**): As described for **7**, 6-amino-3-iodo-4-methoxy-1-(β-d-ribofuranosyl)-1H-pyrazolo[3,4-*d*]pyrimidine **3** (1.7 g, 4.0 mmol) was converted to 1.0 g compound **9** as a white solid in 65% yield with an HPLC purity of 97%. R*_f_* = 0.2 (dichloromethane–methanol = 6:1). UV–vis (MeOH) *λ*_max_: 220, 232, 256 nm; ESI-MS *m*/*z*: 410.1 [M + H]^+^.

*Synthesis of 6-amino-2,5-dihydro-2-(β-d-ribofuranosyl)-4H-pyrazolo[3,4-d]pyrimidin-4-one* (**10**): As described for **7**, 6-amino-4-methoxy-2-(β-d-ribofuranosyl)-2H-pyrazolo[3,4-*d*]pyrimidine **2** (1.0 g, 3.4 mmol) was converted to 650 mg of compound **10** as a white solid in 67.5% yield with an HPLC purity of 98%. R*_f_* = 0.15 (dichloromethane–methanol = 1:1). UV–vis (MeOH) *λ*_max_: 225, 280 nm; ^1^H NMR (DMSO-*d6*, 400 MHz) δ 10.45 (s, 1H, Ar-H), 8.52 (s, 1H, NH), 6.28 (s, 2H, NH_2_), 5.66 (d, *J* = 3.2 Hz, 1H, 1′-H), 5.49 (d, 1H, *J* = 5.2 Hz, 2′-OH), 5.11 (d, *J* = 5.6 Hz, 1H, 3′-OH), 5.01 (t, *J* = 5.2 Hz, 1H, 5′-OH), 4.25–4.35 (m, 1H, 2′-H), 4.08–4.18 (m, 1H, 3′-H), 3.89–3.96 (m, 1H, 4′-H), 3.61–3.64 (m, 1H, CH), 3.46–3.57 (m, 1H, CH); ^13^C NMR (100 MHz, DMSO-*d*6) δ: 160.8 (C(4)), 159.3 (C(6)), 153.7 (C(2)), 128.2 (C(7)), 102.2 (C(5)), 94.0 (C(1′)), 85.1 (C(4′)), 74.7 (C(2′)), 70.0 (C(3′)), 61.3 (C(5′)); ESI-MS *m*/*z*: 284.0 [M + H]^+^.

*Synthesis of 4,6-diamino-2-(β-d-ribofuranosyl)-2H-pyrazolo[3,4-d]pyrimidine* (**11**): As described for **5**, 6-amino-4-methoxy-2-(β-d-ribofuranosyl)-2H-pyrazolo[3,4-*d*]pyrimidine **2** (1.0 g, 3.4 mmol) was converted to 670 mg of compound **11** as a white solid in 70% yield with an HPLC purity of 96%. R*_f_* = 0.15 (dichloromethane–methanol = 10:1). UV–vis (MeOH) *λ*_max_: 225, 262, 302 nm; ^1^H NMR (DMSO-*d6*, 400 MHz) δ 8.42 (s, H, Ar-H), 7.79 (br, 2H, NH_2_), 6.58 (br, 2H, NH_2_), 5.73 (s, 1H, 1′-H), 4.90–5.54 (br, 3H, 2′-OH, 3′-OH, 5′-OH), 4.34 (br, 1H, 2′-H), 4.11 (br, 1H, 3′-H), 3.97 (br, 1H, 4′-H), 3.45–3.74 (m, 2H, CH_2_); ESI-MS *m*/*z*: 283.2 [M + H]^+^.

*Synthesis of 6-amino-4-hydrozino-2-(β-d-ribofuranosyl)-2H-pyrazolo[3,4-d]pyrimidine* (**12**): As described for **6**, 6-amino-4-methoxy-2-(β-d-ribofuranosyl)-2H-pyrazolo[3,4-*d*]pyrimidine **2** (1.0 g, 3.4 mmol) was converted to 695 mg of compound **12** as a white solid in 69% yield with an HPLC purity of 96%. R*_f_* = 0.1 (dichloromethane–methanol = 1:2). UV–vis (MeOH) *λ*_max_: 223, 264, 298 nm; ^1^H NMR (DMSO-*d6*, 400 MHz) δ 8.80, 8.35 (br, 1H, NH), 6.89 (br, H, Ar-H), 6.64 (br, 1H, NH), 5.68–5.84 (m, 1H, 1′-H), 5.38–5.60 (m, 1H, 2′-OH), 4.92–5.24 (m, 2H, 3′-OH, 5′-OH), 4.26–4.43 (m, 1H, 2′-H), 4.04–4.25 (m, 1H, 3′-H), 3.86–4.02 (m, 1H, 4′-H), 3.16–3.70 (m, 5H, CH_2_, NHNH_2_), 3.46–3.57 (m, 1H, CH);ESI-MS *m*/*z*: 298.8 [M + H]^+^. HRMS calcd for C_10_H_16_N_7_O_4_ 298.1264 [M + H]^+^, found 298.1263.

*Synthesis of 6-amino-4-hydroxyamino-2-(β-d-ribofuranosyl)-2H-pyrazolo[3,4-d]pyrimidine* (**13**): Compound **2** (1.0 g, 3.4 mmol) was dissolved in 50% hydroxylamine solution (30 mL, 506 mmol), and the reaction mixture was heated at 60 °C. Upon completion of the reaction as monitored by TLC, the reaction solvent was evaporated under reduced pressure. The resulting precipitate was filtered off and recrystallized from water to yielding 506 mg of compound **13** as a white solid in 50% yield with an HPLC purity of 96%. R*_f_* = 0.2 (dichloromethane–methanol = 1:1). UV–vis (MeOH) *λ*_max_: 223, 285 nm; ^1^H NMR (DMSO-*d6*, 400 MHz) δ 9.97 (s, 1H, NH), 9.86 (s, 1H, OH), 8.16 (s, 1H, Ar-H), 6.81 (br, 2H, NH_2_), 5.59 (d, *J* = 3.6 Hz, 1H, 1′-H), 5.46 (d, 1H, *J* = 5.2 Hz, 2′-OH), 5.14 (d, *J* = 5.2 Hz, 1H, 3′-OH), 4.94–5.08 (m, 1H, 5′-OH), 4.26–4.33 (m, 1H, 2′-H), 4.07–4.16 (m, 1H, 3′-H), 3.87–3.94 (m, 1H, 4′-H), 3.56–3.64 (m, 1H, CH), 3.45–3.53 (m, 1H, CH); ESI-MS *m*/*z*: 298.8 [M + H]^+^, 320.7 [M + Na]^+^. HRMS calcd for C_10_H_14_N_6_O_5_Na 321.0923 [M + Na]^+^, found 321.0920.

*Synthesis of 6-amino-3-(furan-2-yl)-4-methoxy-1-(β-d-ribofuranosyl)-1H-pyrazolo[3,4-d]pyrimidine* (**14**): To a stirred mixture of 6-amino-3-iodo-4-methoxy-1-(β-d-ribofuranosyl)-1H- pyrazolo[3,4-*d*]pyrimidine **3** (600 mg, 1.4 mmol) and bis(triphenylphosphine)-palladium (II) chloride (49 mg, 0.07 mmol, 0.05 eq) in 30 mL of anhydrous DMF was added 2-(tributylstannyl)furan (1.75 g, 4.9 mmol, 3.5 eq) under an nitrogen atmosphere. The reaction mixture was stirred at 90 °C for 18 h and the concentrated to dryness under reduced pressure. The resulting residue was purified by column chromatography to afford 450 mg desired product **14** as a pale-yellow solid in 90% yield with an HPLC purity of 96%. R*_f_* = 0.5 (dichloromethane–methanol = 10:1). ^1^H NMR (DMSO-*d6*, 400 MHz) δ 7.77–7.84 (m, 1H, Ar-H), 7.21 (d, *J* = 3.2 Hz, 1H, Ar-H), 6.96 (s, 2H, NH_2_), 6.61–6.69 (m, 1H, Ar-H), 6.02 (d, 1H, *J* = 4.4 Hz, 1′-H), 5.34 (d, 1H, *J* = 6.0 Hz, 2′-OH), 5.09 (d, *J* = 4.2 Hz, 1H, 3′-OH), 4.76 (t, *J* = 5.6 Hz, 1H, 5′-OH), 4.54–4.61 (m, 1H, 2′-H), 4.17–4.26 (m, 1H, 3′-H), 4.05 (s, 1H, CH_3_), 3.84–3.91 (m, 1H, 4′-H), 3.41–3.64 (m, 2H, CH_2_); ESI-MS *m*/*z*: 364.2 [M + H]^+^. HRMS calcd for C_15_H_18_N_5_O_6_ 364.1257 [M + H]^+^, found 364.1258.

*Synthesis of 6-amino-4-methoxy-3-phenyl-1-(β-d-ribofuranosyl)-1H-pyrazolo[3,4-d]pyrimidine* (**15**): To a stirred mixture of 6-amino-3-iodo-4-methoxy-1-(β-d-ribofuranosyl)-1H-pyrazolo[3,4-*d*] pyrimidine **3** (800 mg, 1.9 mmol), phenylboronic acid (346 mg, 2.8 mmol, 1.5 eq) and potassium carbonate (387 mg, 2.8 mmol, 1.5 eq) in 60 mL of DME–water (2:1) was added Pd(PPh_3_)_4_ (115 mg, 0.09 mmol, 0.05 eq). The reaction mixture was stirred at 90 °C for 4 h. Upon completion of the reaction as monitored by TLC, the mixture was concentrated under reduced pressure. The residue was purified by flash chromatography on a silica gel column to provided 570 mg of product **15** as a white solid in 80% yield with an HPLC purity of 96%. R*_f_* = 0.6 (dichloromethane–methanol = 10:1). ^1^H NMR (DMSO-*d6*, 400 MHz) δ 7.90–8.00 (m, 2H, Ar-H), 7.39-7.54 (m, 3H, Ar-H), 6.92 (s, 2H, NH_2_), 6.07 (d, 1H, *J* = 4.0 Hz, 1′-H), 5.36 (d, 1H, *J* = 5.6 Hz, 2′-OH), 5.08 (d, *J* = 5.6 Hz, 1H, 3′-OH), 4.76 (t, *J* = 6.0 Hz, 1H, 5′-OH), 4.54–4.62 (m, 1H, 2′-H), 4.21–4.31 (m, 1H, 3′-H), 3.99 (s, 3H, CH_3_), 3.85–3.92 (m, 1H, 4′-H), 3.43–3.65 (m, 2H, CH_2_); ESI-MS *m*/*z*: 374.1 [M + H]^+^. HRMS calcd for C_17_H_20_N_5_O_5_ 374.1464 [M + H]^+^, found 374.1461.

*Synthesis of 6-amino-3-ethynyl-4-methoxy-1-(β-d-ribofuranosyl)-1H-pyrazolo[3,4-d]pyrimidine* (**16**): To a stirred mixture of compound **3** (600 mg, 1.42 mmol), Pd (PPh_3_)_4_ (169 mg, 0.14 mmol, 0.1 eq), CuI (53 mg, 0.28 mmol, 0.2 eq) and triethylamine (0.6 mL, 4.26 mmol, 3.0 eq) in DMF (20 mL) was added trimethylsilylacetylene (1.2 mL, 8.52 mmol, 6.0 eq). The reaction mixture was stirred at room temperature overnight. Upon completion of the reaction as monitored by TLC, the mixture was concentrated under reduced pressure. The resulting residue was directly deprotected by treatment with K_2_CO_3_ (39 mg, 0.3 mmol, 0.2 eq) in MeOH (30 mL) at room temperature for 2 h. Upon completion of the reaction as monitored by TLC, the mixture was concentrated under reduced pressure. The residue was purified by flash chromatography on a silica gel column to provided 300 mg of product **16** as a white solid in 65% overall yield with an HPLC purity of 96%. R*_f_* = 0.45 (dichloromethane–methanol = 10:1). ^1^H NMR (DMSO-*d6*, 400 MHz) δ 7.03 (s, 2H, NH_2_), 5.95 (d, 1H, *J* = 4.2 Hz, 1′-H), 5.37 (d, 1H, *J* = 6.0 Hz, 2′-OH), 5.10 (d, *J* = 4.2 Hz, 1H, 3′-OH), 4.75 (t, *J* = 6.0 Hz, 1H, 5′-OH), 4..45–4.55 (m, 2H, 2′-H, CH), 4.10–4.17 (m, 1H, 3′-H), 3.98 (s, 3H, CH_3_), 3.81–3.90 (m, 1H, 4′-H), 3.37–3.58 (m, 2H, CH_2_). HRMS calcd for C_13_H_16_N_5_O_5_ 322.1151 [M + H]^+^, found 322.1150.

*Synthesis of 6-amino-4-methoxy-3-[N-(trifluoroacetyl)aminopropyn-1-yl]-1-(β-d-ribofuranosyl)-1H- pyrazolo[3,4-d]pyrimidine* (**17**): The mixture of compound 3 (500 mg, 1.18 mmol), Pd (PPh_3_)_4_ (145 mg, 0.12 mmol, 0.1 eq), CuI (44.5 mg, 0.25 mmol, 0.2 eq), *N*-(2-propynyl)-2,2,2-trifluoroacetamide (1.07 g, 7.08 mmol, 6.0 eq) and triethylamine (0.5 mL, 3.54 mmol, 3.0 eq) in DMF (20 mL) was stirred at room temperature overnight. Upon completion of the reaction as monitored by TLC, the mixture was concentrated under reduced pressure. The residue was purified by flash chromatography on a silica gel column to provided 310 mg of product 17 as a white solid in 60% yield with an HPLC purity of 95%. R*_f_* = 0.40 (dichloromethane–methanol = 10:1). ^1^H NMR (DMSO-*d6*, 400 MHz) δ 10.16 (s, 1H, NH), 7.03 (s, 2H, NH_2_), 5.95 (d, 1H, *J* = 4.2 Hz, 1′-H), 5.36 (d, 1H, *J* = 6.0 Hz, 2′-OH), 5.10 (d, *J* = 4.2 Hz, 1H, 3′-OH), 4.74 (t, *J* = 5.6 Hz, 1H, 5′-OH), 4.44–4.55 (m, 1H, 2′-H), 4.07–4.17 (m, 1H, 3′-H), 3.96 (s, 3H, CH_3_), 3.76–3.88 (m, 1H, 4′-H), 3.37–3.56 (m, 2H, CH_2_). ESI-MS *m*/*z*: 447.1 [M + H]^+^.

#### 4.2.5. Procedure for Preparation of Glycosylated Products **28**–**29**

*Synthesis of 6-amino-4-methoxy-1-(2-deoxy-3,5-di-(O-p-toluoyl)-α-d-ribofuranosyl)-1H-pyrazolo [3,4-d]pyrimidine* (**28**) *and 6-amino-4-methoxy-1-(2-deoxy-3,5-di-(O-p-toluoyl)-β-d-ribofuranosyl)-1H- pyrazolo[3,4-d]pyrimidine* (**29**): To a stirred suspension of tris[2-(2-methoxyethoxy)ethyl]amine (TDA-1) (411 mg, 1.3 mmol, 0.03 eq) and finely grounded KOH (6.6 g, 118.7 mmol, 2.8 eq) in anhydrous acetonitrile (200 mL) was added 6-amino-4-methoxy-1H-pyrazolo-[3.4-*d*]pyrimidine (purine base A, 7.0 g, 42.4 mmol, 1.0 eq). The reaction mixture was stirred for another 15 min. 2-Deoxy-3,5-di-*O*-(*p*-toluoyl)-α-D-*erythro*-pentfuranosyl chloride (ribose II) (24.7 g, 63.6 mmol, 1.5 eq) was added, and the reaction mixture was stirred at room temperature for an additional 30 min. Upon completion of the reaction as monitored by TLC, the reaction mixture was diluted with ethyl acetate (100 mL) and washed with saturated ammonium chloride solution. The aqueous layer was extracted with ethyl acetate. The combined organic layer was washed with saturated brines solution, dried over anhydrous sodium sulfate, and concentrated under reduced pressure. The resulting residue was purified by column chromatography to afford two products. The compound eluting first was β-isomer 29 (11.4 g, 52%) as a white solid with an HPLC purity of 97.1%. R*_f_* = 0.4 (petroleum ether–ethyl acetate = 3:1). The compound eluting last was α-isomer 28 (2.4 g, 11%) as a white foam with an HPLC purity of 96%. R*_f_* = 0.35 (petroleum ether–ethyl acetate = 3:1).

#### 4.2.6. Procedure for Preparation of Key Intermediates **18** and **19**

*Synthesis of 6-amino-4-methoxy-1-(2-deoxy-α-d-ribofuranosyl)-1H-pyrazolo[3,4-d]pyrimidine* (**18**): As described for **1**, 6-amino-4-methoxy-1-(2-deoxy-3,5-di-(*O*-*p*-toluoyl)-α-d-ribofuranosyl) -1H-pyrazolo[3,4-*d*]pyrimidine **28** (2.4 g, 4.6 mmol) was converted to 1.2 g of compound **18** as a white solid in 89% yield with an HPLC purity of 95%. R*_f_* = 0.35 (dichloromethane–methanol = 10:1). ^1^H NMR (400 MHz, CDCl_3_) δ 7.94 (s, 1H, Ar-H), 6.88 (s, 2H, NH_2_), 6.27–6.45 (m, 1H, 1′-H), 5.60 (d, 1H, *J* = 8.0 Hz, 2′-OH), 4.72 (t, *J* = 5.6 Hz, 1H, 5′-OH), 4.10–4.20 (m, 1H, 3′-H), 3.98 (s, 3H, CH_3_), 3.87–3.95 (m, 1H, 4′-H), 3.35–3.56 (m, 2H, 2′-H) 2.51–2.73 (m, 2H, CH_2_);

*Synthesis of 6-amino-4-methoxy-1-(2-deoxy-β-d-ribofuranosyl)-1H-pyrazolo[3,4-d]pyrimidine* (**19**): As described for **1**, 6-amino-4-methoxy-1-(2-deoxy-3,5-di-(*O*-*p*-toluoyl)-β-d-ribofuranosyl)- 1H-pyrazolo[3,4-*d*] pyrimidine **29** (4.0 g, 7.7 mmol) was converted to 2.0 g of compound **19** as a white solid in 91% yield with an HPLC purity of 97%. R*_f_* = 0.4 (dichloromethane–methanol = 10:1). UV–vis (MeOH) *λ*_max_: 224, 251, 285 nm; ^1^H NMR (DMSO-*d6*, 400 MHz) δ 7.88 (s, 1H, Ar-H), 6.84 (s, 2H, NH_2_), 6.41 (s, 1H, 1′-H), 5.23 (s, 1H, 2′-OH), 4.73 (br, 1H, 5′-OH), 4.39 (br, 1H, 3′-H), 3.97 (s, 3H, CH_3_), 3.78 (s, 1H, 4′-H), 3.41–3.56 (m, 1H, 2′-H) 2.62–3.18 (m, 2H, 2′-H, CH_2_), 2.12–2.23 (m, 1H, CH_2_); ESI-MS *m*/*z*: 282.8 [M + H]^+^.

#### 4.2.7. Procedure for Preparation of Target Compounds **20**–**23**

*Synthesis of 6-amino-1,5-dihydro-1-(2-deoxy-β-d-ribofuranosyl)-4H-pyrazolo[3,4-d]pyrimidin-4-one* (**20**): As described for **7**, 6-amino-4-methoxy-1-(2-deoxy-β-d-ribofuranosyl)-1H-pyrazolo[3,4-*d*] pyrimidine **19** (1.2 g, 4.2 mmol) was converted to 990 mg of compound **20** as a white solid in 88% yield with an HPLC purity of 95%. R*_f_* = 0.15 (dichloromethane–methanol = 5:1). UV–vis (MeOH) *λ*_max_: 257 nm; ESI-MS *m*/*z*: 267.9 [M + H]^+^, 289.9 [M + Na]^+^.

*Synthesis of 4,6-diamino-1-(2-deoxy-β-d-ribofuranosyl)-1H-pyrazolo[3,4-d]pyrimidine* (**21**): As described for **5**, 6-amino-4-methoxy-1-(2-deoxy-β-d-ribofuranosyl)-1H-pyrazolo[3,4-*d*]pyrimidine **19** (1.0 g, 3.5 mmol) was converted to 620 mg of compound **21** as a white solid in 65% yield with an HPLC purity of 96%. R*_f_* = 0.2 (dichloromethane–methanol = 5:1). UV–vis (MeOH) *λ*_max_: 225, 257, 278 nm; ESI-MS *m*/*z*: 267.7 [M + H]^+^.

*Synthesis of 6-amino-4-hydrazino-1-(2-deoxy-β-d-ribofuranosyl)-1H-pyrazolo[3,4-d]pyrimidine* (**22**): As described for **6**, 6-amino-4-methoxy-1-(2-deoxy-β-d-ribofuranosyl)-1H-pyrazolo[3,4-*d*]pyrimidine **19** (1.0 g, 3.5 mmol) was converted to 690 mg of compound **22** as a white solid in 70% yield with an HPLC purity of 96%. R*_f_* = 0.15 (dichloromethane–methanol = 5:1). UV–vis (MeOH) *λ*_max_: 226, 276 nm; ^1^H NMR (DMSO-*d6*, 400 MHz) δ 9.04, 8.49 (br, 1H, NH), 8.09, 7.80 (br, 1H, NH), 6.36 (s, 1H, Ar-H), 6.05, 6.15 (br, 2H, NH_2_), 5.17 (d, 1H, *J* = 4.4 Hz, NH), 4.83 (t, 1H, *J* = 5.6 Hz, 1′-H), 4.45–4.76 (br, 2H, 3′-OH, 5′-OH), 4.33–4.42 (m, 1H, 3′-H), 3.72–3.81 (m, 1H, 2′-H), 3.45–3.55 (m, 1H, CH), 3.35–3.41 (m, 1H, CH), 2.67–2.76 (m, 1H, 2′-H), 2.07–2.17 (m, 1H, 2′-H); ESI-MS *m*/*z*: 282.8 [M + H]^+^.

*Synthesis of 6-amino-4-hydroxyamino-1-(2-deoxy-β-d-ribofuranosyl)-1H-pyrazolo[3,4-d]pyrimidine* (**23**): As described for **13**, 6-amino-4-methoxy-1-(2-deoxy-β-d-ribofuranosyl)-1H-pyrazolo[3,4-*d*]pyrimidine **19** (1.0 g, 3.5 mmol) was converted to 560 mg of compound **23** as a white solid in 56% yield with an HPLC purity of 95%. R*_f_* = 0.2 (dichloromethane–methanol = 4:1). UV–vis (MeOH) *λ*_max_: 223, 276 nm; ^1^H NMR (DMSO-*d6*, 400 MHz) δ 9.75, 9.53 (br, 1H, NH), 7.50 (s, 1H, Ar-H), 6.68 (br, 1H, NH), 6.25 (br, 1H, NH), 5.18 (d, 1H, *J* = 4.4 Hz, OH), 4.78 (t, 1H, *J* = 5.6 Hz, 1′-H), 4.32–4.45 (m, 1H, 3′-H), 3.72–3.85 (m, 1H, 2′-H), 3.35–3.53 (m, 2H, CH_2_), 2.61–2.75 (m, 1H, 2′-H), 2.07–2.17 (m, 1H, 2′-H); ESI-MS *m*/*z*: 283.8 [M + H]^+^.

### 4.3. Proliferation Inhibitory Effect Assay

Cells were plated in a 96-well plate (4–5 × 10^3^ cells per well) in the medium supplemented with 10% fetal bovine serum (FBS) and Dulbecco’s modified Eagle medium (DMEM) at 37 °C with a humidified 5% CO_2_ atmosphere. After 24 h, different concentrations of synthesized compounds (as positive control; 100 μmol/L to 0.01 nmol/L) were added and the cells were exposed to compounds for 72 h. Cell proliferation was evaluated by MTT assay according to the manufacturer’s instructions and incubated for 4 h. The optical density was determined at 570 nm by Varioskan Flash Multimode Reader (Thermo Electron, Finland). The IC_50_ value, which refers to the concentration (μM) of the compound to cause 50% cell death with respect to the control, was calculated according to the inhibition ratios. The experiments were performed in triplicates to obtain the mean cell viability.

## 5. Conclusions

The 8-aza-7-deaza-purine bases **A**, **B**, and **C** were glycosylated with l-*O*-acetyl-2,3,5-tri-*O*-benzoyl-d-ribofuranose (ribose **I**) and 2-deoxy-3,5-di-*O*-(*p*-toluoyl)-α-D-*erythro*-pentafuranosyl chloride (ribose **II**), and further deprotection provided the corresponding key intermediates **1**, **2**, **3**, and **19**. These intermediates were further substituted with NH_2_, OH, NHNH_2_ and NHOH to furnish final products **5**–**13** and **20**–**23**. The iodinated intermediate **3** was cross-coupled with aromatic and alkynyl reagents to provide final products **14**–**17**. The newly synthesized compounds were all well characterized. The structures of *N*^9^- and *N*^8^-glycosylated products were assigned. HMBC analysis of 2D NMR spectra further verified the *N*^9^- and *N*^8^-products. X-ray crystallographic studies unambiguously confirmed the conclusion. Antitumor inhibitory test resulted in some good hits to guide further studies. The protocols discovered herein can be utilized to expand further research in related fields.

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
