# Peer review of "Investigation of 8-Aza-7-Deaza Purine Nucleoside Derivatives"

_molecules, 2019, doi:10.3390/molecules24050983_

Round 1
Reviewer 1 Report
I am satisfied with the authors' answers to my original questions and comments, and therefore, I recommend to accept this manuscript for publication in molecules without any additional changes.
Reviewer 2 Report
This manuscript by Tan have been revised significantly acceding to the previous referees’ comments. I believe the revised manuscript would be publishable in Molecules.
This manuscript is a resubmission of an earlier submission. The following is a list of the peer review reports and author responses from that submission.
Round 1
Reviewer 1 Report
The manuscript describes the synthesis and biological activities of 8-aza-7-deaza purine nucleosides. The authors have developed versatile intermediates for the synthesis of these nucleoside analogues in high diversity. The work have been done carefully and the manuscript is written in a clear and concise manner. Therefore, I would recommend the manuscript be published in Molecules after major revisions suggested below:
1. In page 3, lines 90-94, the authors describe that glycosylation of the purine base A with ribose I gave N9-glycosilted nucleosides 24 or N8-glycosilated nucleoside 25 by tuning the reaction conditions. The authors emphasize nucleoside 25 is thermodynamically stable product. The emphasis will be stronger if the authors can show results of experiments in which compound 24 was treated with TMSOTf in CH2Cl2 at room temperature overnight. The authors are requested to examine these experimental.
2. In Table 2, what is “100μM IC%”? The authors are requested to add these explanations in the text and the table footnotes
3. For Scheme 2, numbering for structure 1 and 3: change “R” to “R1”.
4. For page 3, lines 80 and 82: the sentence is grammatically strange and should be corrected.
5. For Scheme 4: delete “+” near 52% and the lines from ribose II to N8-product, respectively, since N8-product is not observed.
6. Stereochemistry and glycosylation position (N8 vs N9) of products 28 and 29 derived from ribose II is not clearly determined, while the stereochemistry and glycosylation position of products derived from ribose I are carefully determined. The authors are requested to add belief comments about this queries.
Reviewer 2 Report
This paper deals with the synthesis of four 8-aza-7-deaza purine derivatives, usable as intermediates to produce a library of 7-deaza nucleoside analogues.
The English requires careful revision, for example:
Line 28 - “dependent in” should be “dependent on”;
line 34 - “7-substituents” should be “7-substituted analogues”;
line 82 – replace “gave” with “give”;
line 92 – replace “dominate” with “dominant”;
line 116 - replace “treament” with “treatment”;
line 130 - replace “comparing” with “compared”;
line 136 - replace “tells” with “shows”;
line 189 - replace “are” with “is”;
line 192 - delete “was”;
line 197 - replace “yield” with “yields”.
These are only a few examples of grammatical errors.
From the chemistry point of view, the paper is hard to read and follow. For example, at page 3, lines 90–94, the authors write that the glycosylation of riboside I through route b gave compound 25 as the sole glycosylation product, whereas the same route at room temperature gave the N-9 glycosylated analogue 24 as the major product. However, there is no indication of such behaviour in Scheme 1.
I have also found many issues about the structural characterisation of several compounds, in particular for what concerns the mass spectra analysis. At lines 281 and 327, what does the [M]¬+ notation stands for? Having used an ESI source, I do not see how a molecular ion could be formed starting from a neutral molecule (instead of the common pseudomolecular [M+H]+).
The authors reported the HRMS data for some of the compounds. However, the instrument they indicated in the materials and methods (Agilent 6120) is not suitable to achieve the required resolution. Furthermore, the HRMS spectra reported in the Supporting information were recorded using a mass spectrometer equipped with a TOF analyser. The authors should clarify this point.
Finally, from a random control on the correctness of molecular formulas and reported masses, I found several inconsistencies. For example, the author reported for compound 15 the molecular formula C17H20N5O5 (I guess referred to the [M+H]+ pseudomolecular ion), whereas the molecular formula of 15 is C17H21N5O5 (C17H22N5O5 for [M+H]+). The molecular formula of 4 is C10H12BrN5O4, corresponding to MW 345,01 (346,02 for [M+H]+); the authors reported 361.9 and 363, respectively.
The pyrazole rings of almost all compounds are shown as open rings throughout the paper (e.g. see Scheme 1).
Some biological consideration should also be added to the conclusions.
The cited references are overdated. Most of the relevant papers and reviewers published in the last ten years in the fields were not considered.
From other and the above-cited issues, I cannot recommend this manuscript for publication in Molecules.
Reviewer 3 Report
The manuscript entitled “Investigation of 8-aza-7-deaza purine nucleoside derivatives” describes about the synthesis of several derivatives of 8-aza-7-deaza purine nucleoside derivatives, few of their structural confirmation’s studies with 2D-NMR, UV-light, X-ray crystal studies and finally explore about whole series of compounds anticancer activity studies. Authors have good hypothesis of design of novel derivatives of 8-aza-7-deaza purine nucleosides, but synthetic procedures with catalyst (TMSOTf) is already described in ref. 19 (JOC) by other group, good efforts of making but unfortunately not resulted in a great cytotoxicity derivatives. However, is a good study, which might be useful to readers who is working in this area to develop drugs for healthcare.